# The Biological Traumatization of Crops Due to the Enzyme Stage of Enzyme-Mycotic Seed Depletion

**DOI:** 10.3390/pathogens11030376

**Published:** 2022-03-21

**Authors:** Sulukhan K. Temirbekova, Ivan M. Kulikov, Yuliya V. Afanasyeva, Mukhtar Z. Ashirbekov, Olga O. Beloshapkina, Elena A. Kalashnikova, Irina Sardarova, Marat Sh. Begeulov, Dmitry E. Kucher, Natalia E. Ionova, Nazih Y. Rebouh

**Affiliations:** 1All-Russian Research Institute of Phytopathology, Bolshye Vyazyomy, Odintsovo District, 143050 Moscow, Russia; sul20@yandex.ru (S.K.T.); irina.sardarova@gmail.com (I.S.); 2Federal Horticultural Center for Breeding, Agrotechnology and Nursery, 115598 Moscow, Russia; vstisp@vstisp.org (I.M.K.); yuliya_afanaseva_90@bk.ru (Y.V.A.); 3Department of Agronomy and Forestry, Faculty of Agronomy, Manash Kozybaev North Kazakhstan University, 150000 Petropavlovsk, Russia; mukhtar_agro@mail.ru; 4Moscow Timiryazev Agricultural Academy, Russian State Agrarian University, 127550 Moscow, Russia; beloshapkina58@mail.ru (O.O.B.); kalash0407@mail.ru (E.A.K.); mbegeulow@rgau-msha.ru (M.S.B.); 5Department of Environmental Management, Peoples’ Friendship University of Russia (RUDN University), 6 Miklukho-Maklaya Street, 117198 Moscow, Russia; kucher-de@rudn.ru; 6Institute of Fundamental Medicine and Biology, Kazan Federal University, 18 Kremlyovskaya Street, 420008 Kazan, Russia; alekta-meg@list.ru

**Keywords:** fungal pathogens, germination, VIR gene pool, wheat

## Abstract

In the light of Vavilov’s Law, grain traumatization in the standing crop of wheat and other crops due to the enzyme stage of enzyme-mycotic seed depletion (EMSD) was confirmed, the parameters of open and hidden harmfulness were detected, and a scale of plant resistance to such traumatization was developed. The current study demonstrates that pathogen contamination in grains occurs before harvesting and its degree is determined by favorable humidity and temperature conditions and by the open and hidden grain traumatization due to the enzyme stage of EMSD, i.e., the grain’s hydrolytic enzymes providing a growth substrate for a fungal spread that is later substituted by pathogen enzymes leading to grain spoiling and self-warming. The most common technique to preserve grain quality is to support a moisture level that prevents further spreading of the fungi. The grains that are contaminated with very low temperature and humidity levels facilitate the germinability and high quality of the grain. The new ways to withstand EMSD should, first of all, include a selection of activities. Using biological, biochemical and physical (X-ray) methods, genetic sources of resistance towards EMSD were found in the VIR world collection that is recommended for further selection. These sources have become a basis for the varieties, such as Moskovskaya 39, Ilot (winter wheat), Gremme and Gremme 2U (hulless spelt), Alcoran (winter spelt) and Kanysh (spring wheat).

## 1. Introduction

Cereal crops are considered to be one of the most important crops worldwide, which have a major influence on the food security of countries. Nowadays, cereals constitute the basis of our diet, due to the ease of the methods of production, harvesting, storage and transport, the diversity of the geographical areas of production, their richness in constituents of interest nutrition and the diversity of methods of preparation and consumption [1,2]. Grain crops occupy a key important place in the grain production with a significant proportion in the entire agricultural system [3] growing in an area of 724 million/h worldwide [4]. These crops provide nearly 55% of the carbohydrates and daily protein for 85% of the world’s population [5].

However, abiotic stresses, such as frost and rapid temperature changes, are the major factors affecting the growth and yield-related characteristics of grain crops [6,7]. Grain crop yield losses caused by abiotic stresses amount to 40% of world grain crop production [8].This can directly affect the food security of the burgeoning population, which is projected to increase up to 9.1 billion in 2050 [9,10]. Moreover, pests and diseases, particularly fungal diseases, induce highly quantitative and qualitative losses, causing critical damage with significant economic losses [11,12]. Fungal pathogens are very adaptable and can rapidly evolve, even into new resistant strains, depending on the nature of the pathogen, susceptibility of the host, diversity of virulence, density of inoculums and the temperature [13,14].

The phenomenon of EMSD-associated grain traumatization is one of the most harmful phenomena that affects grain crops, particularly the wheat crop. EMSD is a complex pathological phenomenon triggered by abiotic factors, such as excessive humidity and temperature, which are later exacerbated by biotic factors (pathogens) [15]. The etiology and pathogenesis of EMSD includes three stages: the first stage is non-infectious (enzyme), and occurs and develops during the flowering, ripening and harvesting stages when ears and grain are moist; this is followed by the second infectious (mucosal) stage, which is caused by semi-parasitic and saprophytic fungi, mainly species of *Alternaria*, *Helounthosporium*, *Cladosporium*, *Fusarium*, and *Septoria*. At the same time, as a result of non-infectious (enzyme) EMSD processes in the plant, an ideal nutrient substrate is formed for the fungi in the form of the water-soluble products of the hydrolysis of carbohydrates and proteins (sugars and amino acids) released through macro-/microtrauma. When the hydrolytic enzymes of the plant and phytopathogens work together, losses increase, and the sowing, technological and commercial qualities of the grain considerably decreases. The last stage is a special form of the enzyme-mycotic depletion of seeds, which includes the less frequently observed phenomenon in which grain germination during the peak of EMSD-related harvest losses can reach up 30–50% in some years, and the depletion significantly worsens the quality of the grain [16,17]. The 30-year research into EMSD etiology and pathogenesis in wheat, rye, triticale, oat and barley carried out in the former Moscow Department of N. I. Vavilov All-Russian Institute of Plant Genetic Resources (VIR), which is currently the Federal Scientific Center of Horticulture, demonstrated a variety of traumas and their consequences on grains grain, long before harvesting (while blooming and in standing crops) and in windrows [18].

Previous studies demonstrated that the most important ways to increase wheat production is the development of new cultivars [19,20]. Developing new wheat varieties adaptable to the soil and specific climatic conditions in different regions is considered as the most effective option to increase and sustain wheat production [21]. The use of genetic approaches allows for the development of wheat varieties that are tolerant to abiotic (i.e., frost and drought) and biotic stress (, i.e., pathogens), in order to reduce the stress impact on wheat growth and yield [22]. In fact, appropriate selection criteria enable breeders to use the genetic variation for enhancing stress tolerance in crops [23]. The objective of the present study is to investigate the gene pool of winter wheat from the world collection from the N. I. Vavilov Institute of Plant Resources (VIR) to detect the signs of grain traumatization in standing crops and, to select resistant genotypes to produce the winter wheat varieties with group immunity to EMSD.

## 2. Materials and Methods

### 2.1. Experimental Sites

The present study was carried out from 1978–2021 by the Federal Scientific Center of Horticulture (former Moscow Department of VIR); the phytopathological research by the All-Russian Scientific Research Institute of Phytopathology and Russian State Agrarian University; the Moscow Timiryazev Agricultural Academy (from 1978 to 1995 by the Siberian Scientific Research Institute of Agricultural Chemicalization (Krasnoobsk, Novosibirsk Region)); and the pass surveys by the VIR Experimental Station in Dagestan (2015–2021).

### 2.2. Plant Materials

Two thousand samples and recognized varieties of the gene pool of wheat, rye, barley and oat from the VIR world collection were investigated. The evaluated samples were obtained from Scandinavia, Western Europe, the U.S.A., Canada, CIS countries and the Russian Federation.

### 2.3. Climatic Conditions and Soil Characteristics

The climate of the Moscow region is moderately humid and continental. The average annual amount of precipitation ranges from 450–800 mm. The cumulative temperatures above 10 °C reduce from 2100 °C in the south-east and the east to 1900 °C in the north-west, reducing the vegetation period (above +10 °C) from 140–145 to 120–125 days.

The region’s water and thermal availability make it an appropriate territory for growing all typical temperate crops. The Stupinsky District is classified as the second (II) agroclimatic region and is located at the center of the region, being a part of subdistrict 11a with sod-podzolic clay-loam soils.

In winter, they freeze down to 50–75 cm in open areas, and down to 30–50 cm in protected ones. The soil de-freezes from 21–29 April and reaches its physical maturity on average on 20 May (clay-loam soils) and on 18 May (sandy-loam soils). The duration of the frost-free season varies from 120 to 135 days, which is long enough for the cultivated crops to reach their full ripeness. The seasonal snow cover forms by 25 November/2 December; its average height is 35 cm. The cover sustains for 137–143 days. The Hydro-Thermal Coefficient (HTC) varies between 1.3 and 1.4.

### 2.4. Methodology

The collections of winter wheat and rye, spring barley and oat (500 seeds/m^2^) were planted at the optimum times between 25 and 27 August (winter varieties) and between 1 and 3 May (spring varieties), as part of scientific crop rotation. For planting, an SSFK-7M planting machine was applied in a plot area of 2 m^2^. NPK 68-60-30 was used as a pre-planting material, and N 50 was used as a plant food in spring. The used farming techniques were typical for the region.

The wheat collection was investigated following the VIR Methodological Recommendation [24,25] and CMEA Wide Unified Classifier for *Triticum* L. [26]. The grains’ resistance to EMSD was estimated by applying original techniques [26].

The anatomical and pathomorphological studies were conducted using an ordinary scanning electron microscope and X-ray (Family PRDU, Figure 1) techniques to estimate the degree of the grains’ contamination and resistance to EMSD [27,28]. The X-ray diffraction analysis of the seeds was carried out by the Agrophysical Research Institute, and the informative X-ray diffraction images of the plant seeds are explained in Figure 2 and Figure 3. The technique of microfocus radiography, in comparison to the traditionally used contact radiography, allows for the attainment of X-ray images of seeds with a projected image magnification of up to several dozen times.

Enzyme electrophoresis was applied to estimate the grains’ protein stability. The study was performed during the firm–ripe stage and while storing using a modified B.B.-O. Gromova technique [29]. The grains’ sowing characteristics and germinability were estimated as required by GOSTs 12038–84 and 12039–66 [30].

## 3. Results and Discussion

It can be observed that the complex lesions (traumatizations) of the grain crops were determined by their etiological groups. The first group mainly included the mechanical lesions the grains obtained while being harvested, transported and processed. The second group mainly included the traumas, such as cell turgescence, which occurred in grains due to different EMSD factors, which we have termed biological traumatization.

These lesions occurred in standing crops and were provoked by the EMSD enzyme stage. Initially, they were only found in the winter wheat grown in the Moscow and Kursk regions and at the VIR Experimental Station in Dagestan, but, later, they were also discovered in triticale, rye, corn, barley, oat, buckwheat, pea crops, coleseed bearers, and later in the spring, wheat, rye and barley in Western Siberia, making them an excellent demonstration of Vavilov’s Law [31].

The third group combined complex and mixed lesions, in which minor biological damages were facilitated by the mechanical effects of transportation, threshing and drying. The investigations carried out in the Moscow and Kursk regions at the VIR Experimental Station in Dagestan and Western Siberia from 1978 to 2021, demonstrated that biological traumatization can be open and hidden. Such images allow us to visualize the fundamentally smaller details of the seed structure, which slightly differ in density (Figure 2 and Figure 3).

In other words, the biological traumatization is not only a “gate” for different pathogens to enter the a grain, but also a way to attract them through the availability of the feeding substrate that is formed by hydraulic enzymes and penetrates through microfractures, which has a direct effect on the grain quality.

### 3.1. Hidden Biological Traumatization during Blooming

Having assessed a gene pool of more than two thousand samples using the original methods [8], we developed the conclusion that humidity affects the plants in the way they lost their dry matter (DM) at every stage of development. What is more, DM losses occurred not only in the grain during its milk, wax and full ripeness, but even earlier, during the plant’s blooming phase, i.e., the hydrolytic enzymes (amylases and proteases) decomposing available biopolymers became active during this very stage. Hydrolyzing the biopolymers of the plants’ vegetative organs is considered to be one of the reasons for a seed’s bad set in humid weather, despite profuse pollination. Having entered a grain, a pathogen adds its own enzymes to the ongoing hydrolysis, additionally producing toxic (and non-toxic) metabolites and remaining hidden inside a flower (see Figure 4).

Among the winter wheat varieties investigated, four groups in relation to DM loss were registered [32]. It was the milk ripeness phase that turned out to be the most sensitive to EMSD. When moisturizing the ears and grains during the blooming, wax and full ripeness phases, the DM loss was not as high as the milk ripeness. However, the genotypical characteristics of the varieties in the groups did not change. Additionally, the investigation results for the years 2012–2013 were analyzed.

The agricultural and weather conditions for the years 2012–2013, favored the estimation and selection of the samples from the crop gene pool to meet the region’s limitation factors, such as winterhardiness and the progressing biological (enzyme-assisted) and mycotic traumatization, due to EMSD. A total of 1200 samples were analyzed to find those with the highest level of winterhardiness and yield capacity (Table 1).

In the samples suffering from significant biological traumatization in standing crops as well as from early and grain blight, the crops exposing the highest and lowest resistance to EMSD, such as Ivanovskaya 16 k-58526 (Ivanovo region), Moskovskaya 39 k-64160 (Moscow Research Institute of Agriculture), Bassard k-64027 (Germany), Gelderseris b/k (Netherlands), Zarya k-49916 (Moscow Research Institute of Agriculture), and Ibis k-45335 (Germany), were selected. These varieties were resistant to both the enzyme and mycotic stages of EMSD. The varieties, such as Moskovskaya 56 b/k, Nemchnovskaya 24 b/k (Moscow Research Institute of Agriculture), Bersy k-64013 (The Netherlands), Zentos k-64030 (Switzerland), Obelisk k-62052 (Germany), Varmalands k-34230 (Sweden), and Orestis k-64034 (Germany) exposed either a relative or moderate resistance to EMSD. The varieties, such as Mironovskaya 808 k-43920, Standard (Ukraine), Kazanskaya 285 k-63560 (Tatar Research Institute of Agriculture), Nika Kubany k-63404, Umanka k-6304 (Krasnodar Region), and Yantarnaya 50 k-54610 (Moscow Research Institute of Agriculture), demonstrated a high tolerance to EMSD. Simultaneously, Polykarlik 3 k-54508 (Ukraine), Faktor k-64028, Fakta k-57582 (Germany), Lives k-63016 (Finland), Expert k-63273 (Austria), Zadorinka k-63119 (Irkutsk Region), and Nike k-64051 (Poland) turned out to be susceptible to EMSD (Table 1).

The selected samples resistant to biological traumatization in terms of DM loss (resistance groups I–III) formed a proper 1000-grain mass and produced a sufficient yield within the studied region.

The grain analysis of the selected crops is presented in Table 2.

(a) A good proportion of properly shaped, large, glassy grains of 39.6–48.0 g in the Moskovskaya 56, Nemchnovskaya 24, Zarya, Gelderseris, Bassard, Ivanovskaya 16, Moskovskaya 39, Mironovskaya 808, Nika Kubani, and Kazanskaya 285 varieties. At the same time, susceptible varieties, such as Polukarlik 3, Fakta, Faktor, Lives, Expert, Nike and Zadorinka, had hydrolyzed grains, some of which had signs of biological traumatization, i.e., fractures with Alternaria/Fusarium mycelium on their surfaces. Their 1000-grain weight varied from 34.1 to 38.1 g, and the yield, from 170 to 240 g/m^2^.

(b) The samples included in resistance groups I, II and III showed the proper regenerative capacity of the root system after wintering and were resistant to snow mold (Microdochium nivale). The varieties, such as Gelderseris, Bassard, Zentos, Obelisk, and Orestis, demonstrated complex resistance and were almost unaffected by brown rust and powdery mildew, so we recommend them for further selections.

### 3.2. Open and Hidden Biological Traumatization during Grain Formation and the Milk/Wax Ripeness Phases

Excessive humidity due to rain and abundant dew has a negative effect on the weak tissues of a just-formed grain kernel, making it susceptible to biological traumatization. This effect that reveals itself during grain formation and the milk/wax ripeness phases may be both open and hidden and has been observed not only in wheat, rye and triticale, but also in barley, oat, corn and buckwheat. In humid weather conditions, as the kernel’s volume increases, so does its inner hydrostatic pressure, which stretches and rips its elastic shell. Sometimes, a kernel resembles an inflated balloon with its crease fractured (Figure 5) and oozing enzyme-decomposed biopolymers.

Sugar release and osmotic moisture increase the pressure inside the kernel, squeezing out the products of hydrolysis through the blossom part of the crease as well as through the microfractures of the endosperm and the pores of the pappus, in which the products are concentrated to ooze as “honeydew” (Figure 6). A similar phenomenon was observed not only in wheat, but also in rye, triticale, oat, barley, pulses (peas) and the seed bearers of crucifers [32,33].

The hydrolysis products are immediately infected by *Alternaria*, *Helminthosporium*, *Cladosporium*, *Septoria*, and *Fusarium*. However, if the weather becomes dryer, hydrolytic enzymes stop working, the biopolymers stop oozing and the traumas are healed, locking the infection inside the kernels and grains. Sometimes, in the case of long-term EMSD, the inner tissues of kernels and grains completely hydrolase, oozing through the fractures and pores, so such kernels and grains become corrugated and rotten and lose all their goodness (Figure 4). Similar phenomena were registered (Figure 7) in 1978, 1979, 1980, 1981, 1985, 1986, 1987, 1989, 1991, 1994, 1995, 2003, 2008, 2013 as well as in 2018, when in July and August, the front of moisture in soils exceeded the long-term average value by 3–4 times.

### 3.3. Open and Hidden Biological Traumatization during the Full Ripeness Phase

The following types of biological grain traumatization were identified for the full ripeness phase: very weak, weak, average and strong. These kinds of biological traumatization were detected in wheat (Figure 8), and later in triticale, rye, barley, buckwheat and peas.

Most of the traumatized grains were severely infected with fungi, both inside and outside. Around the fractures, brown and pink mycotic spots were formed. All the types of traumatization led to a reduced germinating power and worsened both the laboratory and field germinability of the seeds. When planted, the traumatized seeds reduced the yield capacity of the Mironovskaya 808 variety by 30.1–36.1% (in 1983, 1988 and 2004) and by 69.1–73.8% (in 1984, 2006 and 2013), when compared to the healthy planting material.

The hidden traumatization caused by EMSD is analogous to that in the phase of open grain traumatization during the full ripeness phase. Starch grains have “gnawed edges” and erosion (Figure 9).

The hidden fractures become an open gate for soil and drop infections to enter the grain (Figure 10).

The acting EMSD phytopathogenic factor, in this case, is determined by the composition of pathogenic agents typical for a particular territory. The factor increases grain depletion, and therefore, even in case of insignificant precipitation during the full ripeness phase, results in the grain being traumatized.

In commercial batches, such grains are similar to healthy ones, but their microscopic examination reveals fractures on their surface and a cavern inside. These processes probably occur while sorting wet grains undergo the first non-infectious enzyme stage of EMSD when biopolymers are degraded by hydraulic enzymes and squeezed out through the fractures in the shell.

Figure 11, Figure 12 and Figure 13 demonstrate the caverned grains of wheat, rye and barley. Similar processes were observed in oat, triticale, pea and buckwheat grains. This symptom is typical in all the regions in Russia, as well as in Germany and the Netherlands.

To detect the deeply hidden biological traumatization in grains, the X-ray method has proved promising. An analysis of the X-ray images of the grain demonstrated that the first enzyme stage of EMSD could be easily detected as lengthwise, dark lines with blurred edges on the bright background of healthy grain tissues.

The X-ray images of enzyme erosions were different from those of mechanical damages. Unlike the latter, the former was wide and had blurred edges to reflect endosperm mass loss along shell raptures. The mechanical damages, in this respect, could be not observed at all, since the loss of tissue due to mechanical shell raptures was too insufficient to affect the optical density of the image. The enzyme erosions in the crease area also had blurred edges.

During the milk/wax ripeness phases, the fractures of the grain provoked by rain and abundant dew may quickly heal, thanks to good weather that prevents further biopolymer oozing and locks the infection inside. Analyzing the grains grown in the Nonchernozem zone of Western Siberia, the VIR Experimental Station in Dagestan and the Kursk Region in humid years during their blooming, plumpness and ripeness periods, showed they were affected by a large number of pathogens, including the most commonly found *Alternaria alternata* (Fr.) Keissl. and *Cladosporium herbarum* Lk. Ex Fr., *Bipolaris sorokiniana Shoem*., *Fusarium avenaceum* (Fr.), Sacc, *F. oxysporum* Schlecht, *F. culmorum* (W.G.Sm) Sacc, and *Septoria nodorum* (Berk.) Berk. When planted after five-year storage, such seeds demonstrated much worse germinability and resilience, if compared to resistant varieties.

It was found [32] that a combination of certain weather temperatures and humidity levels caused EMSD that destroyed the structure of proteins and starch in wheat plants. This occurred due to the increasing activity of (α + β) amylases and proteases that, in some very susceptible varieties, led to the fact that no total protein stain could be found, although such an investigation requires the same amount of protein that is determined spectrophotometrically at 260 and 280 nm.

Other negative consequences included the reduced activity and even disappearance of particular isoenzymes, and changes of the protein spectrum that manifested itself in the reduced representation of certain electrophoretic fractions or their complete decomposition. It is noteworthy that the obtained changes in the amylase spectrum corresponded to those in the protein spectrum. The resistant varieties (Zarya, Ibis, Bassard, and Bersy) suppressed the proteolytic activity intensity better than the susceptible ones.

Suppressing the proteolytic activity, the resistant varieties preserved the cell-organelle integrity and contained the negative effect of EMSD. The data on protein destruction and amylase inhibitor activity enabled us to detect different degrees of plant adaptation at the level of protease activity during the EMSD enzyme stage.

Thus, the wheat varieties resistant to EMSD are characterized by anabolic processes, i.e., stable protein synthesis. On the other hand, in the plants susceptible to EMSD, the metabolic processes shift to become catabolic, and are characterized by excessive protein hydrolysis. The wheat varieties, whose response is an increased activity of hydrolytic enzyme protease, amylase and a structural and conformal protein rearrangement to react to abiotic factors (high weather temperature and humidity) are to be described as susceptible.

The data of proteino- and enzymograms showed that the grains of wheat, triticale, rye, barley and oat are affected by excessive humidity, while plumpness could neither restore their biochemical parameters by the full ripeness phase nor after 6 months or 5 years after harvesting (during storage). We pointed out [32] that the disrupted structures of proteins and enzymes affected the germinability and resilience of the seeds. After a 5-year storage period, the grains of the varieties susceptible to EMSD, such as Faktor, Fakta, lives, Expert, Nike, Zadorinka and Polykarlik 3, which were harvested in the humid year of 2013, reduced their field germination rate by 44.0–57.0 %, when compared to their initial rate in 2013, while their resilience reduced by 15.0–41.0% (Table 2).

In the resistant and relatively resistant varieties, such as Gelderseris, Bassard, Moskovskaya 39, Ivanovskaya 16, Moskovskaya 56, Ibis, Bersy, Zentos, Obelisk, and Orestis, the field germinability comprised 50–70%, i.e., it reduced by 25–39.0%.

The varieties included in resistance groups I and II preserved their resilience at 55.0–75.0%. Their Alternaria and Fusarium infection rate varied within 52.0–15.0%, while that for the susceptible varieties varied within 52.0–100%. Here, it should be noted that before being sent to storage, their Alternaria infection rate did not exceed 12.0–30.0%, and that of Fusarium—12.0–35.0%, which can be explained by the slightest change in humidity and temperature, where storing activates breezing processes and hydrolytic enzymes (amylase and protease) as well as the pathogens inside and outside a grain. Sometimes the amount of mico- and microflora increases exponentially. The described factors are the main reason grains lose their germinability and resilience while being stored (Figure 14).

Therefore, it becomes apparent that the pathogenic infection of the grains occurs before and not after their harvesting, as has been pointed out by several researchers [33]. The infection rate in a batch of grain depends on favorable humidity and weather temperature as well as on the presence of open or hidden grain traumatization that has occurred due to the EMSD enzyme stage, i.e., when a grain’s hydrolytic enzymes produce feed substrate for fungi, which, in turn, enhance its destruction resulting in grain spoiling and self-warming. Our research demonstrated that the joint activity of grain and fungus enzymes exponentially increases the number of the latter.

The most common method for sustaining the quality of grain and seeds is to maintain storage humidity at such a level to prevent the further development of the fungi that a grain contains. In other words, low temperature and humidity enhances grain resilience and quality. They also prevent moisture transfer within a bulk grain [34].

Developing techniques to fight EMSD should first rely upon the selection process. Using biological, biochemical and physical (X-ray) methods enabled us to detect the genetic sources of resistance to EMSD that can be recommended for implementation. They have become a basis for the varieties, such as Moskovskaya 39, Ilot (winter wheat), Gremme, Gremme 2U (spelt), Alkoran (winter spelt), and Kanysh (spring wheat).

A 9-point scale was used to account for the resistance to biological traumatization in EMSD-invaded grains, in which EMSD symptoms are assessed from 0 to 4. The number “0” in traumatized plants [35] corresponds to “9” in resistant plants (no traumatization). The number “4” in traumatized plants corresponds to “1” (severe traumatization up to 100%).

The scale enables one to estimate a plant’s resistance to EMSD after rain, in standing crops, and before harvesting to predict the yield loss and estimate the genotypical disposition of a variety to the second (infectious/mycotic) EMSD stage (Alternaria spot, cladosporiosis, Septoria spot, Fusarium head blight), which is especially important for regional and perspective varieties [31].

The most valuable varieties detected in this way may be additionally tested using the artificial infection method to confirm their resistance to the above-mentioned pathogens.

Vavilov’s Law is a great discovery of significant practical value for biology.

## 4. Conclusion

Based on the principles of Vavilov’s Law, we determined the phenomenon of biological traumatization in standing crops as the result of the EMSD enzyme stage, not only in wheat but also in other crops. This biological traumatization of open and hidden types can be assessed using the resistance scale we developed. Currently, we are carrying out research aimed at healing and mitigating the consequences of biological traumatization from which plants suffer in wet weather.

The results of our perennial studies are presented in Figure 14 and devoted to the long-term storage of the gene pool of different crops from the VIR world collection and reserve stocks. This confirmed the endopathic cause of germinability and resilience loss in seeds due to excessive humidity, and the development of enzymic and mycotic infections while grains are in the blooming/plumping stages. Phytopathologists are recommended to apply these biochemical criteria for the estimation of crop resistance to ear and grain diseases.

## Figures and Tables

**Figure 1 pathogens-11-00376-f001:**
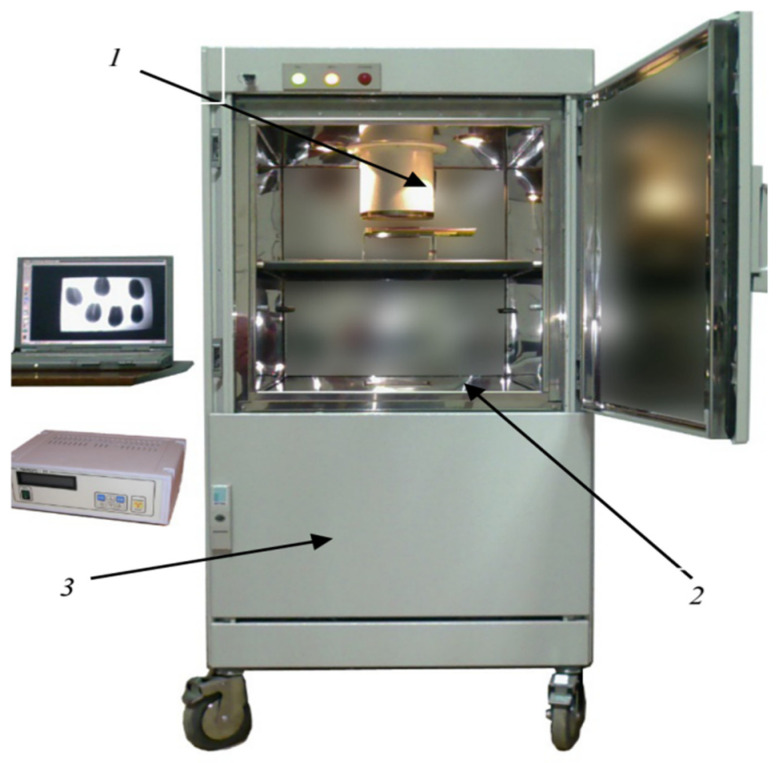
Family PRDU: 1. X-ray source; 2. digital image receiver; and 3. specialized camera for X-ray work.

**Figure 2 pathogens-11-00376-f002:**
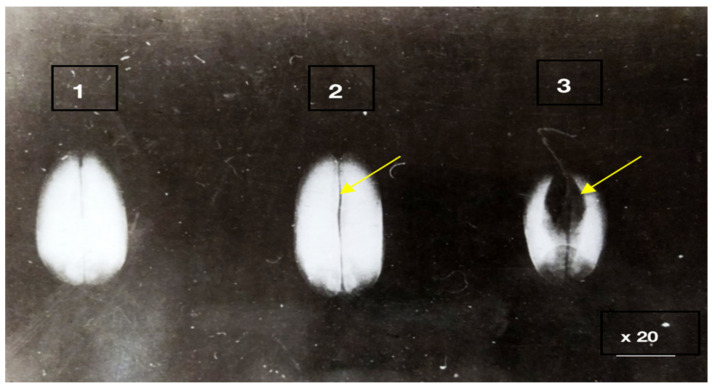
X-ray diffraction analysis of winter wheat seeds. 1. Control seeds without damage. 2. A grain with biological traumatization in the spinal part. 3. A symptom of a dent on a grain with biological traumatization.

**Figure 3 pathogens-11-00376-f003:**
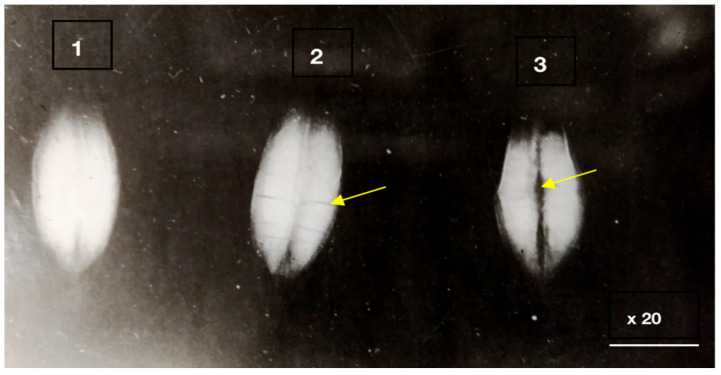
X-ray diffraction analysis of sprout seed traumatization. 1. Control seeds without damage. 2. A grain with biological traumatization in the spinal part (sprout seed). 3. A symptom of a dent on a grain with biological traumatization.

**Figure 4 pathogens-11-00376-f004:**
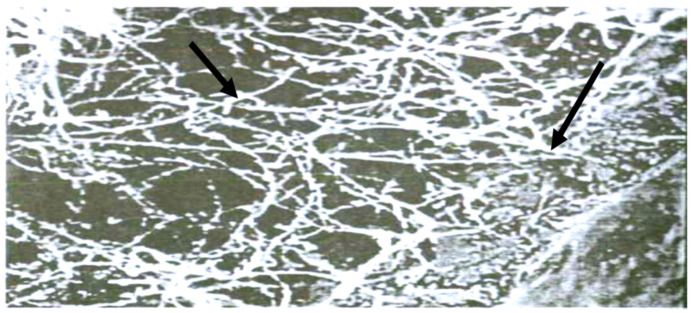
Hidden biological traumatization in a wheat flower ovary due to the EMSD enzyme stage (×800).

**Figure 5 pathogens-11-00376-f005:**
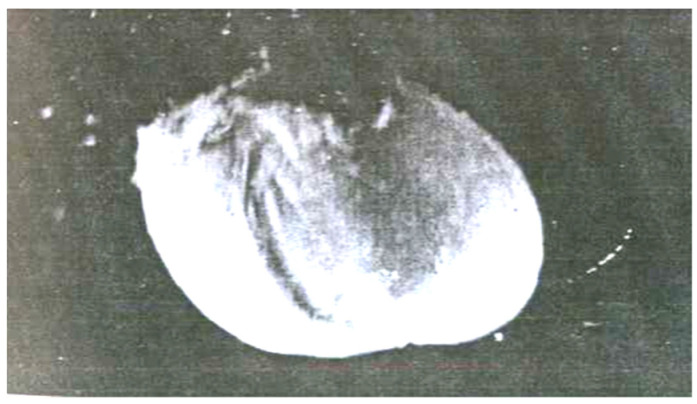
Balloon-shaped kernel (×30).

**Figure 6 pathogens-11-00376-f006:**
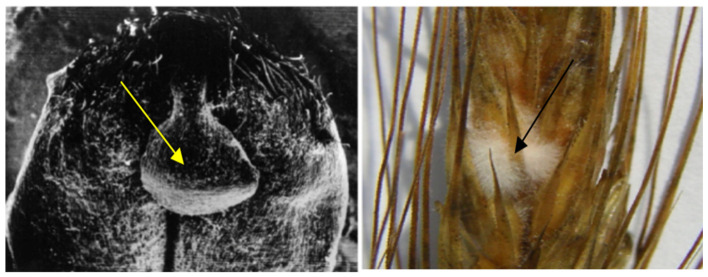
“Honeydew” (secretion) formed under the influence of the enzyme stage of EMSD in the milk ripeness phase of wheat (**left**) (×50). Phytopathogen *Fusarium* spp. on the “honeydew” (**right**) (×10).

**Figure 7 pathogens-11-00376-f007:**
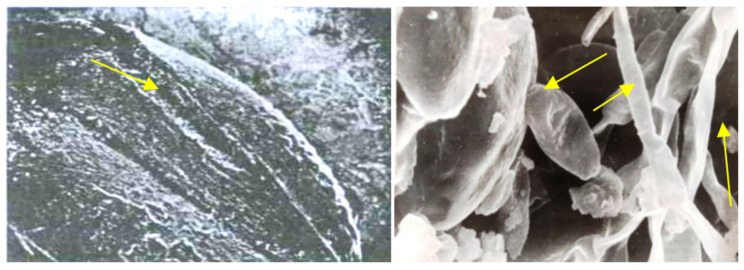
Opening fracture in wheat grain (**left**) (×40). Introduction of phytopathogenic infection after biological traumatization (×4500), (**right**).

**Figure 8 pathogens-11-00376-f008:**
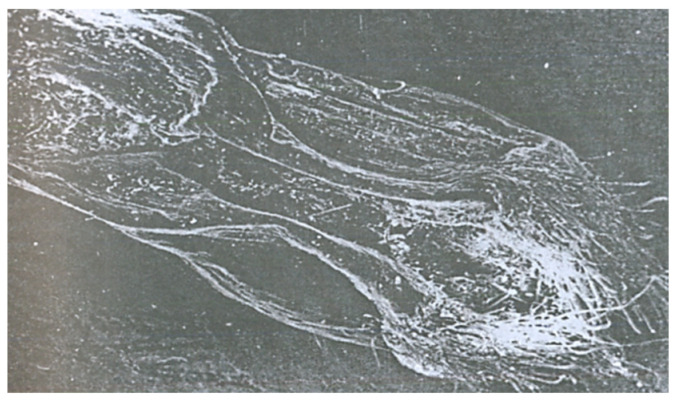
Oozed wheat grain (×30).

**Figure 9 pathogens-11-00376-f009:**
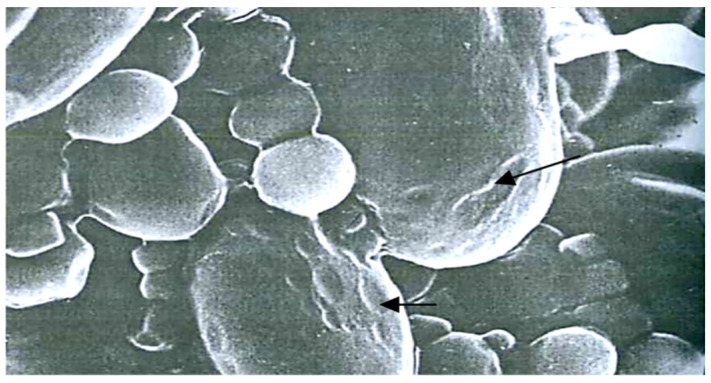
“Gnawed edges” and erosion of starch grains (×5000).

**Figure 10 pathogens-11-00376-f010:**
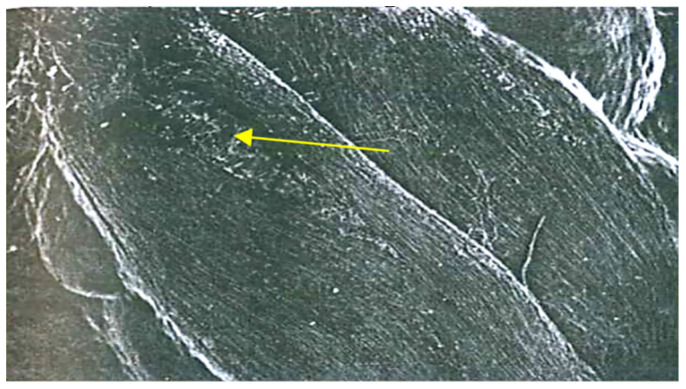
Hidden biological traumatization in standing crops during the full ripeness phase (×30).

**Figure 11 pathogens-11-00376-f011:**
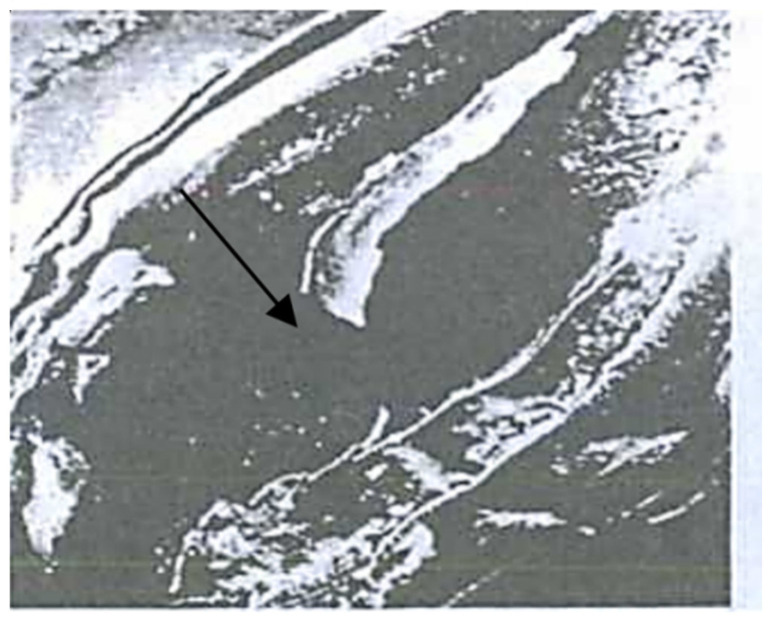
Hidden biological traumatization of a wheat grain due to the EMSD enzyme stage that causes a cavern to form (×45).

**Figure 12 pathogens-11-00376-f012:**
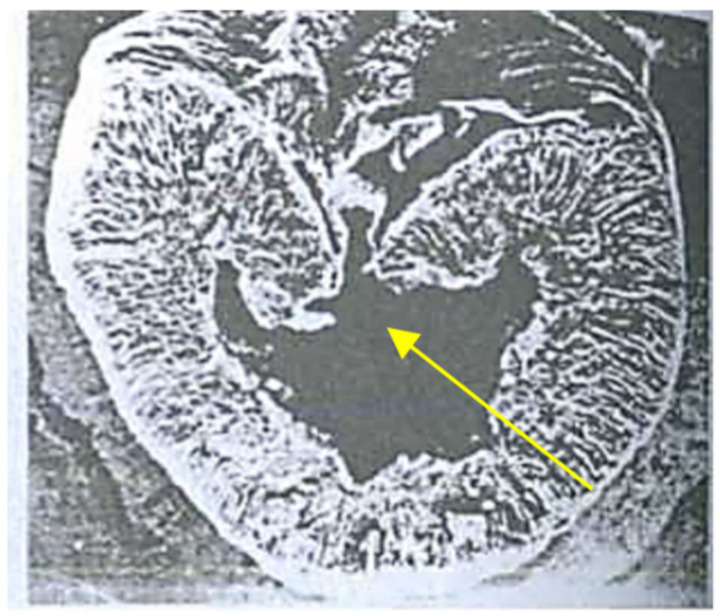
Hidden biological traumatization of a barley grain due to the EMSD enzyme stage that causes a cavern to form (×45).

**Figure 13 pathogens-11-00376-f013:**
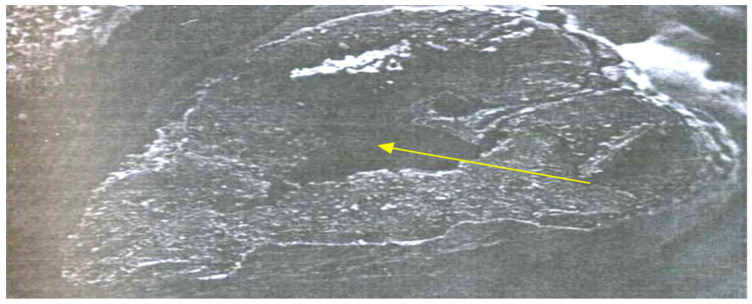
Hidden biological traumatization of a rye grain (×100).

**Figure 14 pathogens-11-00376-f014:**
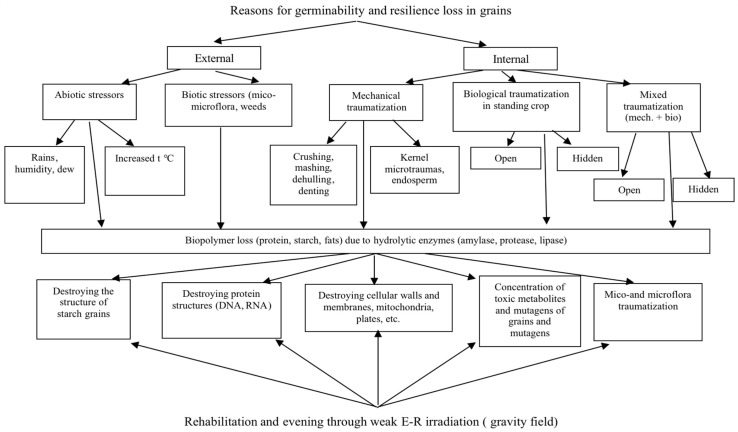
Rehabilitation and evening through weak E-R irradiation.

**Table 1 pathogens-11-00376-t001:** Winter hardiness and productivity of the selected winter wheat samples in terms of their resistance to EMSD during a humid 2013 and following 5-year storage.

№ Registration VIR	Cultivars	Origins	Winter Hardiness, Point	Weight per 1000 grains, g	Yield, g/m^2^
Dry matter losses up to 15%, stability of group I: stable					
45335	Ibis	Germany	5	34.9	260
49916	Zarya	Russia	7	39.6	350
-	Gelderseris	Netherlands	8	40.4	425
64027	Bassard	Germany	8	41.3	400
64160	Moskovskaya 39	Russia	7	45.7	430
58526	Ivanovskaya 16	Ivanovo region	9	41.0	367
Losses of dry matter up to 25%, stability of group II: relatively stable					
65760	Moskovskaya 56	Russia	7	48.0	400
65757	Nemchnovskaya 24	Russia	5	41.7	450
64013	Bersy	Netherlands	7	39.0	290
64030	Zentos	Switzerland	5	37.5	280
62052	Obelisk	Germany	5	38.2	300
64034	Orestis	Germany	5	37.4	310
34230	Varmalands	Sweden	7	39.3	302
Losses of dry matter up to 35%, stability of group III: highly tolerant					
43920	Mironovskaya 808	Ukraine	9	40.7	380
54610	Yantarnaya 50	Russia	7	38.9	310
63560	Kazanskaya 285	Russia	7	39.8	320
63041	Umanka	Russia	7	38.7	300
63404	Nika Kuban	Russia	7	39.1	340
Dry matter losses of more than 35%, stability of group IV: unstable (susceptible)					
54508	Polokarlik 3	Ukraine	5	34.1	220
57582	Fakta	Germany	5	33.4	210
64028	Faktor	Germany	5	35.6	180
63016	lives	Finland	5	36.1	200
63273	Expert	Austria	4	35.4	185
64051	Nike	Poland	5	36.5	170
63119	Zadorinka	Irkutsk region	5	38.1	240
LSD_05_				0.8	90

**Table 2 pathogens-11-00376-t002:** Quality of the winter wheat seeds affected by excessive humidity after a 5-year storage period.

Cultivars	Germination, %			Survivability %, 2018 г.	Infestation Rate, %			
					2013 г.		2018 г.	
	2013	2018	Difference		*Altemaria altematа*	*Fusarium* spp.	*Altemaria altemata*	*Fusarium* spp.
Resistance to EMSD								
Ibis	92.0	64.0	28.0	67.0	7.0	3.0	25.0	10.0
Zarya	94.0	60.0	34.0	71.0	12.0	4.0	28.0	12.0
Gelderseris	90.0	61.0	29.0	69.0	10.0	5.0	30.0	10.0
Bussard	95.0	58.0	37.0	70.0	8.0	3.0	20.0	7.0
Ivanovskaya 16	90.0	62.0	28.0	73.0	15.0	8.0	31.0	10.0
Moskovskaya 39	92.0	67.0	25.0	72.0	12.0	5.0	15.0	10.0
Relatively resistant to EMSD								
Moskovskaya 56 cт.	95.0	70.0	25.0	75.0	7.0	2.0	21.0	12.0
Bersy	90.0	57.0	33.0	64.0	12.0	5.0	28.0	10.0
Zentos	90.0	55.0	35.0	60.0	8.0	4.0	51.0	10.0
Obelisk	89.0	59.0	30.0	67.0	10.0	7.0	49.0	15.0
Orestis	85.0	50.0	35.0	62.0	15.0	10.0	52.0	15.0
Varmalands	87.0	48.0	39.0	55.0	10.0	5.0	46.0	15.0
Highly tolerant to EMSD								
Mironovskaya 808 sт.	85.0	52.0	33.0	61.0	12.0	8.0	50.0	22.0
Nika Kuban	90.0	43.0	47.0	53.0	10.0	12.0	45.0	18.0
Kazanskaya 285	80.0	45.0	35.0	55.0	15.0	10.0	57.0	25.0
Umanka	85.0	47.0	38.0	54.0	9.0	15.0	40.0	30.0
Not resistant to EMSD								
Polokarilik 3	87.0	35.0	52.0	37.0	12.0	25.0	52.0	100
Fakta	85.0	34.0	51.0	33.0	20.0	28.0	56.0	100
Faktor	87.0	30.0	57.0	34.0	20.0	18.0	60.0	100
lives	80.0	36.0	44.0	39.0	10.0	27.0	85.0	90.0
Expert	85.0	33.0	52.0	41.0	30.0	25.0	80.0	100
Nike	80.0	31.0	49.0	35.0	25.0	18.0	90.0	100
Zadorinka	65.0	17.0	48.0	15.0	20.0	35.0	60.0	100

## Data Availability

Not applicable.

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
