# Peer review of "The Biological Traumatization of Crops Due to the Enzyme Stage of Enzyme-Mycotic Seed Depletion"

_pathogens, 2022, doi:10.3390/pathogens11030376_

Round 1
Reviewer 1 Report
The paper is really interesting, however it is difficult to state the type of that since it is a sort of mixture od hypotheses and experimental search.
Biological degradation of any crop is a known phenomenon for millenia in the history of the human race. The hypothesis focuses on one of the possible reasons and authors make attempts to find evidence. Actually the remoistening and the temporal factors of fungal development has been written by several authors. The enzyme stage determination may be a novel component in this process.
I suggest the authors to focus on the exact materials and methods rather than raising hypothetic versions. Also, in the conclusions the research results are to be discussed rather than plylosophic postulates.
The paper would need a thorough grammatic improvement, - a native reader would make a benefit in this case, Furthermore uniform use of varietal names should be welcome since the various uses of Latin and Cyrillic aphabets may cause disorientation.
Author Response
Please, find below the attached file.

Reviewer 2 Report
GENERAL COMMENTS
This study represent an application of the Vavilov’s law to the phenomenon of EMSD-associated grain traumatization in standing wheat crop, and fits perfectly within the scope of “pathogens”. This is a complex phenomenon that is pathologically activated by excess humidity and temperature (abiotic factors). Consequently, improving our understanding of this process is a pressing task due to the uncertainties that limit the secure predictions of productivity of many crops.
In my opinion, this research accomplishes the objectives set out. The authors have gathered a large database (30 years), performed in the present Federal Scientific Center of Horticulture (and its precedent and associated Institutions), where pioneering results were also obtained in this matter. These data correspond to varieties of the gene pool of wheat, rye, barley and oat from the VIR world collection. Samples came from Scandinavia, Western Europe, USA, Canada, CIS countries and Russian Federation. Collections were planted in the optimum times (May and August), according to varieties, each one in a plot area of 2 m2, and soil was treated with farming techniques typical of the region.
Briefly, the wheat collection was investigated following VIR Methodological Recommendation and CMEA Wide Unified Classifier for Triticum L. The grains’ resistance to EMSD was estimated by applying original techniques, as indicated. Analyses have been performed by SEM, to estimate the degree of the grains’ inner contamination and resistance to EMSD, and Enzyme electrophoresis was applied to estimate the grains’ protein stability. Finally, the grains’ sowing characteristics and germinability were estimated as it is required by the GOSTs indicated.
It seems that research has been well conducted. They have applied appropriate methods, obtaining consistent results. The article is quite well structured, clearly explained, correctly illustrated, and acceptably written in English.
However, I have some concerns, which could be included in the final version.
SOME CONCERNS AND QUESTIONS
1-Ls91-93. The authors should be much more specific when indicate that “… were carried out using an ordinary scanning electron microscope and X-ray techniques …”: there are many SEM and X-ray techniques, and they must also specify the devices used, the working conditions, sample coating, as well as the place where said facilities were located. Additionally, the type of samples used may require special treatments.
The authors also indicate “X-ray techniques”. Do they mean EDX (Energy Dispersive X-Ray Analysis)? (EDX is an X-ray technique used to identify the elemental composition of materials).
2-Ls258-263. The authors do not show any image obtained with X-Ray technique. Why? It can be read in this paragraph several details which cannot be observed. These images would be a very valuable complement. However, the indicated EDX does not give this type of images. What X-Ray techniques the authors refer to? Please, clarify this point.
3-I suggest subdividing the section 2 (Material and Methods) in three subsections. For instance: 2.1. General setting; 2.2. Material; 2.3. Methodology. This subdivision (or similar) could be a way to facilitate, for example, the reading or to prepare research protocols.
4-Figures inserted in the text are an ideal complement to show, understand and follow the indications given. However, all they should be notably improved. In general, all these images need a graphical scale and should be contrasted. In particular, the Fig. 9 is “burned”. In addition, the authors should be more explicit, explaining briefly and indicating, for instance with arrows, some interesting aspects for the reader.
5-The section “References” should also be notably improved:
I recommend increasing the number of references: this revision suggests some of them.
Please, give preference to international cites and, if possible, universal methods.
There is excessive auto-citation: the first author represent nine from fourteen references.
Please, write also in rectangular brackets the order numbers of the list.
SPECIFIC COMMENTS
Ls36-42. The repercussion of the works of Nikolay I. Vavilov in this investigation, and his universal itinerary, should place him in a relevant place in the references list, as in fact you have already done in the Introduction. I encourage authors to cite some article by this author on the indicated lines to put him in this relevant place.
L58. I suggest writing at the beginning of this sentence “The present study was …”. However the time span indicated seems excessive. Please, revise.
L66. Please, write the first letter of “western” in uppercase.
L80. The term “hydrothermal” is also common in Geology. Therefore, it is better write “The hydrothermal coefficient (HTC, Selyaninov, 1928)…” for avoid confusions. Please, see this reference at the end of this revision.
L85. Typing mistake. The reader should assume that the authors mean “2 m2” (in superscript). Other possibility it would be “2 × 2 m”. Please, revise.
L97. I advise giving a reference for GOSTs. Please, see this reference at the end of this revision, although may be incomplete.
Ls119,182. I advise insert an upper blank line to avoid confusion.
L198. Please, revise this sentence.
L200. Perhaps a typing mistake: “… now only in wheat but also in rye, …”. Please, improve and change “now” to “not”.
L215. I believe it sound wrong to say “…the precipitation depth in July …”. It would be preferable say “… in July and August the front of moisture in soils exceeded …”
Ls215,373. We can define perennial as “lasting or existing for a long or apparently infinite time; enduring or continually recurring”. I think that this is not the meaning of the two sentences of the text. Please, revise.
L216. I advise inserting an upper blank line to avoid confusion.
L226. Please, close the bracket open: “… (in 1983 …”
L235. I believe that the authors should better write “phase”, instead of “case” because they refer to the title of the section mentioned above (L216).
L246. Typing mistake: “burley” should be “barley”
L266. Please, revise the use of “shall” in this sentence.
L302. I advise writing “We have pointed out [8] that ...”, or similar.
L327. I advise insert two additional articles in the reference. The comment is not very positive and should be softened in some way.
Fig. 10. The letters in this graphic are very tiny. Please, improve.
L366. Typing mistake. Please, delete the number “5”.
REFERENCES CITED IN THIS REVISION
Selyaninov G.T., 1928. About climate agricultural estimation. Proceedings on Agricultural Meteorology 20, 165–177.
GOST 12038-84. Seeds of agricultural crops. Methods of analysis. SB. GOSTs. Moscow: Publishing, Standards, 2004. 47 с.
Author Response
Please, find below the attached file.

Round 2
Reviewer 1 Report
I accept the corrections done by the authors.
Author Response
Dear Reviewer!
All comments have been taken into consideration and corrected in the manuscript.
Best Regards,
Dr. Rebouh